# Field Study of the Effects of Two Different Environmental Conditions on Wheat Productivity and Chlorophyll Fluorescence Induction (OJIP) Parameters

**Valentina Spanic [1],\*** , **Selma Mlinaric [2]** , **Zvonimir Zdunic [1]** and **Zorana Katanic [2]**

[1] Agricultural Institute Osijek, Juzno Predgradje 17, 31000 Osijek, Croatia; zvonimir.zdunic@poljinos.hr

[2] Department of Biology, Josip Juraj Strossmayer University of Osijek, Cara Hadrijana 8a, 31000 Osijek, Croatia; selma.mlinaric@biologija.unios.hr (S.M.); zorana.katanic@biologija.unios.hr (Z.K.)

\* Correspondence: valentina.spanic@poljinos.hr; Tel.: +385-31-515-569

**Abstract:** Wheat is one of the main cereal crops for ensuring food supply. Thus, increasing grain yield is a major target for plant breeders, where insights into wheat productivity can be gained by studying the activity of the components of photosynthetic apparatus. The objectives of this study were to evaluate the agronomical performance of three winter wheat varieties and test photosynthetic efficiency over two different locations. Chlorophyll fluorescence was used to evaluate the maximum quantum yield of photosystem II (PSII) ($TR_0/ABS$) and performance index on absorption basis ($PI_{abs}$) of flag leaves and glumes of heads at the flowering stage until the mid-senescence stage. The grain yield of all varieties on average was significantly higher at Osijek compared to Tovarnik. Variety Tika Taka exhibited the highest yield reduction (27.1%) at Tovarnik compared to Osijek, followed by El Nino (20.5%) and Vulkan (18.7%), respectively. A higher amount of precipitation in June at Tovarnik provoked higher Fusarium head blight disease intensity, which could be seen as the bleaching of plant heads at the plots and resulted in an earlier decrease in photosynthetic activity. Therefore, earlier senescence and contracted grain fill duration could occur.

**Keywords:** abiotic stress; chlorophyll fluorescence; grain yield; wheat

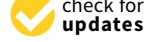

## 1. Introduction

Grain growth and senescence of plants are influenced by many factors during the vegetative period, including climatic conditions and varietal types [1,2]. Climate change leads to increased occurrence of extreme events, such as drought or heavy precipitation [3]. Therefore, biotic and abiotic stresses affect crop yield worldwide [4], causing a decline in crop production and food safety. Thus, to ensure global food demand, there is a need to increase crop yields [5]. One way to increase wheat yields is by improving photosynthetic efficiency. It was reported [6] that the productivity of agricultural plants is directly associated with their photosynthetic activity, which is of high importance for agronomy as the most likely route toward enhanced biomass production [7]. Crop yield is determined by leaf photosynthesis, which contributes to carbohydrate accumulation [8]. Accordingly, the study of photosynthetic activity is very important as an integrated approach to analyzing the physiological and biochemical bases of stress tolerance. It is believed that wheat heads and flag leaves of wheat play a major role as sources of assimilates during grain filling. The duration of photosynthetic functionality of leaves is closely correlated to the grain yield of wheat [9]. Therefore, flag leaves of wheat are the major photosynthetic organs supplying the grain of wheat. Ear photosynthesis makes a significant contribution to the grain yield of bread wheat, from 13% to 33% in the absence of stress [10]. Furthermore, grain yield, yield components, and ratio of grain weight/leaf area are positively related to the contribution of ear photosynthesis [11]. Previously, it was concluded that under stress conditions, wheat heads have a higher contribution to grain yield [12]. According to previous research [13], photosynthesis contribution in wheat heads to grain yield was 20%.

When all tissues and organs of the parental plant die, resources from senescing organs are remobilized to the developing grains [14]. The organs closest to the developing grains (flag leaves and glumes in the heads) will generally senescence last [15]. The processes involved in senescence are important because they occur during grain filling, and evidence suggests that early senescence may be yield limiting [16]. Delayed senescence, or the 'stay-green' trait, is associated with extending photosynthetic duration, potentially increasing the availability of resources for grain filling. Accordingly, the most important agronomic traits such as nutrient use efficiency, yield, and quality are influenced by senescence timing [17]. 'Stay-green' varieties with late senescence often, but not always, exhibit higher grain yields. Higher grain protein concentration can be associated with early and/or efficient nutrient remobilization [18]. The grain weight, which is also regulated throughout the reproductive stage, is affected by sink limitation and, to a lesser extent, by the capacity and duration of photosynthesis and mobilization of assimilates to grain [19]. Therefore, senescence accelerates in the presence of biotic or abiotic stresses. Fungal pathogens provoke changes in metabolic pathways, such as photosynthesis [20], but fungi can also induce effects such as tissue necrosis, structural damage, and stomatal closure on its hosts, which inhibit photosynthesis [21].

Chlorophyll (*Chl*) fluorescence is one of the frequently used methods in monitoring and screening for stress tolerance in plants due to the high sensitivity of plant responses to the alterations induced on the photosynthetic system, especially photosystem II (PSII) [22,23]. Photosynthesis is particularly sensitive to environmental factors [24], making photosynthetic measurements an important component of plant stress studies. Various biophysical parameters derived from *Chl* fluorescence transient measurements can help to understand the energy flow through PSII as a highly sensitive signature of photosynthesis and provide useful indicators of stressful conditions [25]. Fast *Chl* fluorescence transient measurements with high time resolution provide a noninvasive and rapid method to study PSII activity changes. Therefore, photosynthetic parameters are a good tool to screen crop varieties for conditions of climate change [26]. The site of electron transfer within PSII is sensitive to environmental stress, particularly heat and moisture stress [27]. Modification of the redox status can cause an imbalance in cellular energy and result in photosynthetic apparatus response due to environmental stress sensors [28]. In addition, most plants adapt themselves to water stress by dissipating excess excitation energy thermally with the downregulation of PSII activity to protect the photosynthetic apparatus from photodamaging effects under water stress [29]. Abiotic stresses primarily reduce the photosynthetic efficiency of plants, due to their negative consequences on chlorophyll biosynthesis, performance of the photosystems, electron transport mechanisms, gas exchange parameters, and many others [28]. Additionally, changes in metabolic pathways such as photosynthesis occur during the infection of plant tissue with fungal pathogens [20]. The objectives of this study were to evaluate the agronomical performance of Croatian winter wheat varieties in two different environments and test the photosynthetic performance in flag leaves and glumes of the heads using chlorophyll *a* fluorescence and two main chlorophyll fluorescence induction (OJIP) parameters, namely $TR_0/ABS$ and $PI_{abs}$. This is particularly interesting from the point that the two locations used for agronomical and OJIP parameter measurements in this study have different climatic conditions.

## 2. Materials and Methods

### 2.1. Field Experiments

Three winter wheat (*Triticum aestivum* L.) varieties were planted during the vegetative season 2019/2020 at two experimental locations (Osijek (45°27′ N, 18°48′ E) and Tovarnik (45°10′ N, 19°09′ E)) in randomized complete block design with two replications. Plots were $7 \times 8$ m rows with 0.20 m row spacing, and sowing density was adjusted to 330 g m$^{-2}$. For the present study, winter wheat varieties from Agricultural Institute Osijek were selected: Tika Taka (82 cm tall, medium duration, high yielding with good productive tillering), Vulkan (78 cm tall, medium duration, high yielding with stable quality), and El Nino

(73 cm tall, medium duration with good assimilates translocation in the grain) (https://cdn.poljinos.hr/upload/documents/Katalog%20psenica_2020_%20web%20(2).pdf, accessed on 10 September 2021).

To meet the winter wheat plant nutrient requirements, fertilization was performed (N:P:K 120:80:120 kg ha$^{-1}$). Pesticides and herbicides were utilized as necessary to minimize the effects of pests and weeds. During the vegetative period, the sum of precipitation at Osijek and Tovarnik amounted to 408.6 and 448.3 mm, respectively (Figure 1a,b). The average annual temperatures at Osijek and Tovarnik were 11.1 and 11.7 °C, respectively. The grain yield was measured by harvesting the whole area of each plot, corrected to 14% moisture (on a wet basis) and converted into dt ha$^{-1}$. Measures of density (test weight in kg hl$^{-1}$) and 1000 kernel weight (g) were obtained by GAC 2100 (DICKEY-john) and the MARVIN grain analyzer, respectively. Chlorophyll *a* fluorescence was monitored at 3, 10, 18, and 26 days after flowering (DAF) at the two locations.

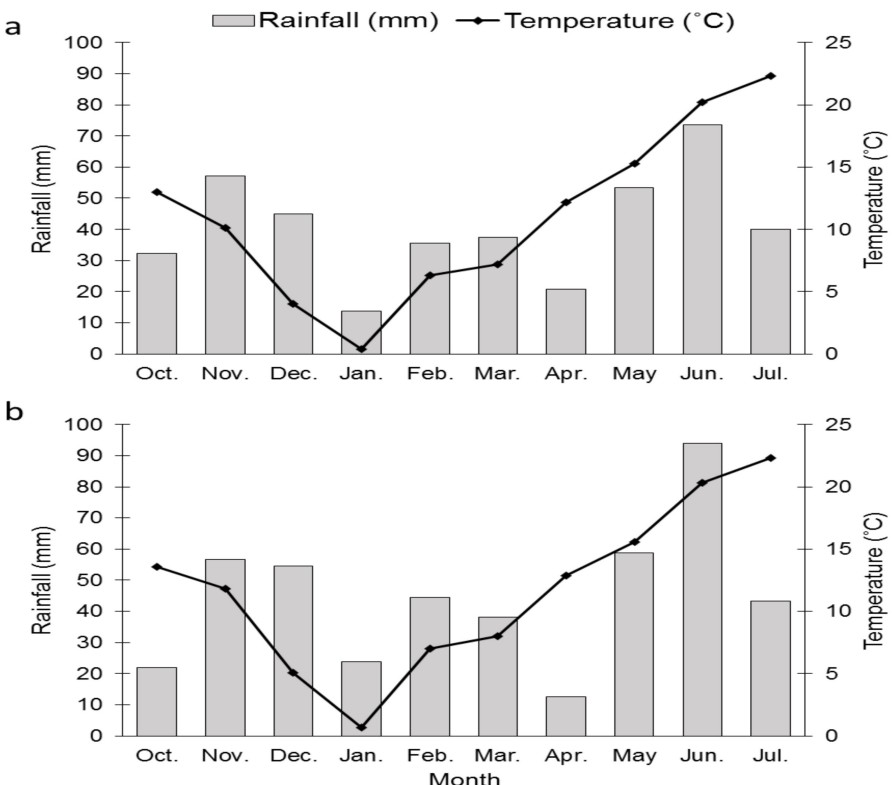

**Figure 1.** Climate diagrams for vegetative season 2019/2020 at Osijek (**a**) and Tovarnik (**b**), Croatia.

## 2.2. Fast Chlorophyll a Fluorescence Measurements

Fast chlorophyll *a* fluorescence transient of ten randomly selected wheat flag leaves and heads per plot was measured using the Plant Efficiency Analyzer (Handy PEA, Hansatech, UK) at each measurement point (3, 10, 18, and 26 days after flowering). A lightweight leaf clip with a shutter plate was used for the dark adaptation of flag leaves and glumes of plant heads in the field. After 30 min of dark adaptation, the chlorophyll *a* fluorescence transient was induced with a saturated red-light pulse (3200 µmol m$^{-2}$ s$^{-1}$, peak at 650 nm), after which data were recorded over a period of 1 s. To calculate biophysical parameters that quantify the stepwise energy flow, the OJIP test was used [30]. Parameters analyzed in this study were the maximum quantum yield of PSII (TR$_0$/ABS) and performance index on absorption basis (PI$_{abs}$) [31].

## 2.3. Statistical Analysis

Measured OJIP test parameters, TR$_0$/ABS and PI$_{abs}$, were presented as mean values of twenty flag leaves/heads ± standard deviation. Statistica 12 software was used to

perform analysis of variance (ANOVA), followed by Fisher's LSD test to detect significant differences between means at a significance level of $p < 0.05$. for each variety separately and at each location.

## 3. Results

### 3.1. Grain Yield and Other Agronomical Traits

The analysis of variance (ANOVA) demonstrated significant effects ($p < 0.001$) of location on the grain yield and test weight of three winter wheat varieties, while 1000 kernel weight significantly differed between varieties (Table 1). The grain yield of varieties was significantly lower at Tovarnik compared to Osijek (Table 2). Based on the climatic data (Figure 1a,b), the unfavorable cropping location due to optimal conditions for Fusarium head blight disease was Tovarnik with a mean grain yield equal to 93.33 dt ha$^{-1}$. At Osijek, wheat varieties yielded 119.98 dt ha$^{-1}$ on average. Grain yield was lower by 18.7, 20.5, and 27.1% in Vulkan, El Nino, and Tika Taka, respectively, at Tovarnik compared to Osijek. Test weight was significantly higher at Tovarnik compared to Osijek. The 1000 kernel weight was not significantly different between the two locations (Table 2). Overall, the values of mean grain yield varied from 91.1 dt ha$^{-1}$ (Tika Taka) at Tovarnik to 125.0 dt ha$^{-1}$ for the same variety at Osijek (Figure 2a). The mean rankings based on the mean test weight demonstrated that the variety El Nino had the best ranking with the highest test weight (84.5 kg hl$^{-1}$) compared to Tika Taka and Vulkan, respectively, at Osijek (Figure 2b). At Tovarnik, Vulkan had the highest value of test weight (85.7 kg hl$^{-1}$) (Figure 2b). At both locations, 1000 kernel weight in Tika Taka (52.84 g at Osijek and 51.94 g at Tovarnik) was higher than those in El Nino and Vulkan. At both locations, the lowest 1000 kernel weight was observed for variety Vulkan (34.43 g and 38.65 g, respectively) (Figure 2c). In general, all wheat varieties harvested at Osijek had higher 1000 kernel weight with a larger number of grains than those harvested at Tovarnik. Furthermore, sterility of upper and lower spikelets was observed at Tovarnik (Figure 3).

**Table 1.** Analysis of variance for grain yield and agronomical components.

| Source of Variation | DF | MS | | |
|---|---|---|---|---|
| | | GY | TW | TKW |
| Variety | 2 | 15.7 | 0.26 | 287.74 *** |
| Location | 1 | 2131.3 *** | 8.84 *** | 10.70 |
| Replication | 1 | 439.1 ** | 0.01 | 0.04 |
| Error | 7 | 27.40 | 0.37 | 3.47 |

***, ** = significant at $p < 0.001$ and 0.01, respectively; DF—degrees of freedom, MS—mean square, GY—grain yield, TW—test weight, TKW—1000 kernel weight.

**Table 2.** Grain yield and agronomical components at different locations.

| Location | GY (dt ha$^{-1}$) | TW (kg hl$^{-1}$) | TKW (g) |
|---|---|---|---|
| Osijek | 119.98a | 83.67b | 43.66a |
| Tovarnik | 93.33b | 85.38a | 41.77a |

GY—grain yield, TW—test weight, TKW—1000 kernel weight; different lowercase letters represent significantly different values between locations ($p < 0.05$).

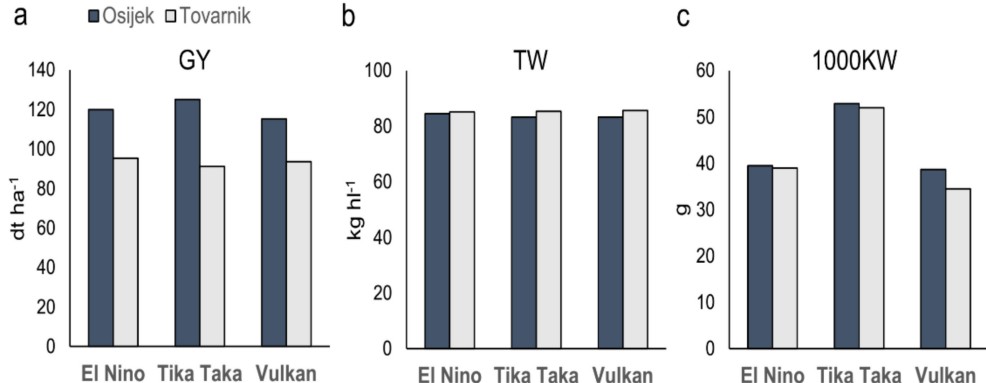

**Figure 2.** Grain yield (GY) (dt ha$^{-1}$) (**a**), test weight (TW) (kg hl$^{-1}$) (**b**), and 1000 kernel weight (1000 KW) (g) (**c**) in varieties El Nino, Tika Taka, and Vulkan at two locations (Osijek and Tovarnik).

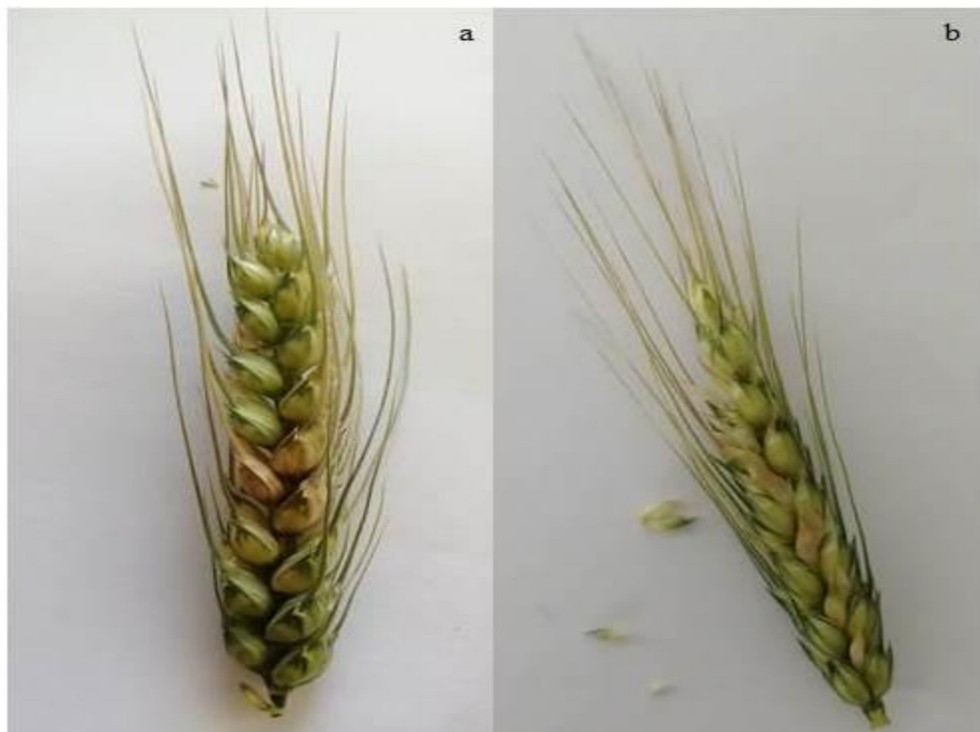

**Figure 3.** Heads of Vulkan at 26 DAF at Osijek (**a**) and Tovarnik (**b**).

### 3.2. Chlorophyll a Fluorescence Measurements

The ANOVA demonstrated significant effects of different plant tissue (PT), location (L), variety (V), and measurement time (MT) on the analyzed OJIP test parameters ($p < 0.001$). Interactions PT*L, PT*V, and L*V did not show a significant effect on PI$_{abs}$. Interactions L*MT, L*V*MT, and PT*L*V*MT did not have a significant effect on TR$_0$/ABS, while interactions PT*L*V and L*V*MT did not have a significant effect on PI$_{abs}$. Interactions PT*MT, V*MT, PT*L*MT, and PT*V*MT had a significant effect on both OJIP test parameters (Table 3).

**Table 3.** Analysis of variance for two chlorophyll fluorescence induction (OJIP) parameters.

| Source of Variation | DF | MS | |
|:---:|:---:|:---:|:---:|
| | | $TR_0/ABS$ | $PI_{abs}$ |
| PT | 1 | 0.3790 *** | 203.460 *** |
| L | 1 | 0.0320 *** | 12.802 *** |
| V | 2 | 0.0194 *** | 15.273 *** |
| MT | 3 | 0.0400 *** | 54.942 *** |
| PT*L | 1 | 0.0177 *** | 0.073 |
| PT*V | 2 | 0.0064 *** | 0.535 |
| L*V | 2 | 0.0020 *** | 0.003 |
| PT*MT | 3 | 0.0049 *** | 3.594 *** |
| L*MT | 3 | 0.0005 | 2.027 *** |
| V*MT | 6 | 0.0030 *** | 0.843 *** |
| PT*L*V | 2 | 0.0013 *** | 0.350 |
| PT*L*MT | 3 | 0.0008 * | 3.492 *** |
| PT*V*MT | 6 | 0.0023 *** | 0.986 *** |
| L*V*MT | 6 | 0.0002 | 0.302 |
| PT*L*V*MT | 6 | 0.0005 | 0.522 * |
| Error | 913 | 0.0003 | 0.223 |

***, * = significant at $p < 0.001$, 0.05; DF—degrees of freedom, MS—mean square, V—variety, PT—plant tissue, L—location, MT—measurement time.

### 3.2.1. Maximum Quantum Yield of Primary Photochemistry ($TR_0/ABS$) and Performance Index on Absorption Basis ($PI_{abs}$) in the Flag Leaves

In the flag leaves of El Nino at Osijek, the maximum quantum yield of primary photochemistry ($TR_0/ABS$) was significantly lower at the third measurement (18 DAF) compared to the first (3 DAF) and second measurements (10 DAF). At Osijek, variety Tika Taka, after an initial increase at 10 DAF compared to 3 DAF, significantly decreased $TR_0/ABS$ at 18 DAF. Vulkan showed a significant decrease at the last measurement (26 DAF) compared to 3 and 10 DAF. Furthermore, a significant decrease in $TR_0/ABS$ in Tika Taka occurred at 26 DAF compared to 18 DAF. At Tovarnik in El Nino, after a significant increase at 10 DAF compared to 3 DAF, a significant decrease in $TR_0/ABS$ occurred at the third measurement (18 DAF). In Tika Taka and El Nino at Tovarnik, a significant decrease occurred at 26 DAF compared to previous measurements (3, 10, and 18 DAF). In Vulkan, a significant decrease occurred at the third measurement (18 DAF) compared to the first measurement (3 DAF) and continued to decrease at 26 DAF (Figure 4a–f).

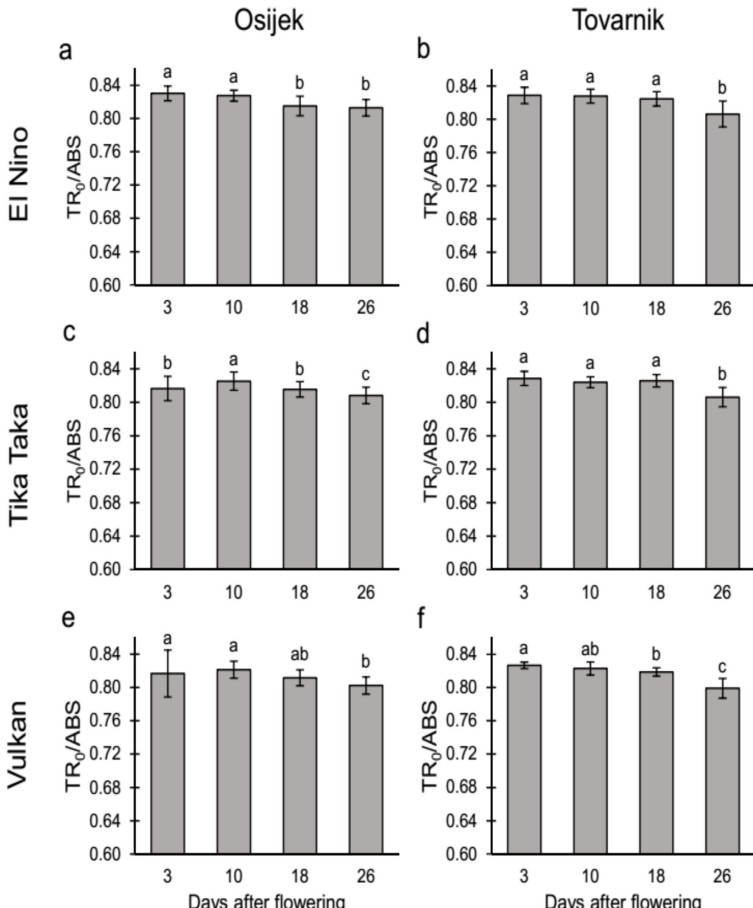

**Figure 4.** Maximum quantum yield of primary photochemistry (TRo/ABS) presented as mean value ± standard deviation of 20 flag leaves of variety El Nino (**a**,**b**), Tika Taka (**c**,**d**), and Vulkan (**e**,**f**) through four measurement points at two locations separately (Osijek and Tovarnik). Standard deviations are presented as vertical bars. Lower-case letters above columns are used to present results of statistical data analysis obtained by Fisher's LSD test and point to statistically significant difference ($p < 0.05$) between values of TRo/ABS measured at 3, 10, 18, and 26 days after flowering for each wheat variety separately and at each location.

In the flag leaves of El Nino at Osijek, a significant decrease in performance index ($PI_{abs}$) occurred at 18 DAF compared to the first measurement (3 DAF). In Tika Taka at Osijek, a significant decrease occurred at 18 DAF compared to 10 DAF and continued to decrease at 26 DAF. Vulkan significantly decreased $PI_{abs}$ already at the second measurement (10 DAF) at Osijek, and a further decrease occurred at 26 DAF. At Tovarnik in El Nino, a significant decrease in $PI_{abs}$ occurred at 26 DAF compared to 3, 10, and 18 DAF. In Vulkan, a significant decrease occurred at 18 DAF compared to 10 DAF, with a further decrease at 26 DAF. At Tovarnik, Tika Taka significantly decreased $PI_{abs}$ at the last measurement (26 DAF) compared to previous measurements (3, 10, and 18 DAF) (Figure 5a–f).

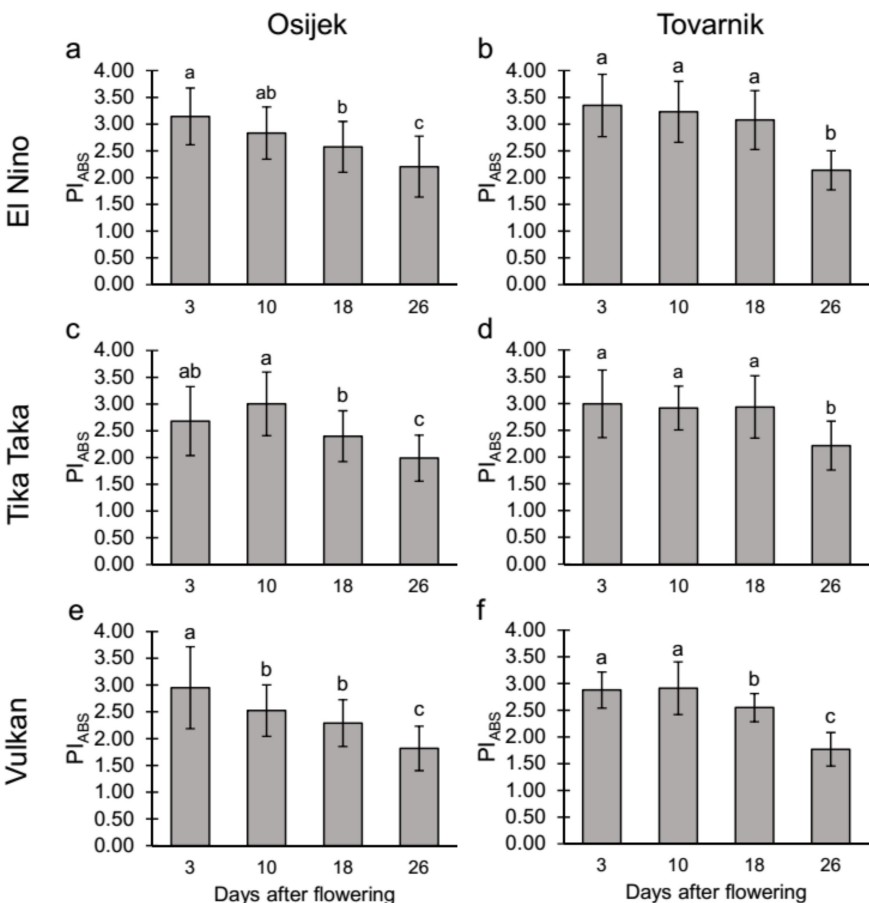

**Figure 5.** Performance index ($PI_{abs}$) presented as mean value ± standard deviation of 20 flag leaves of variety El Nino (**a,b**), Tika Taka (**c,d**), and Vulkan (**e,f**) through four measurement points at two locations separately (Osijek and Tovarnik). Standard deviations are presented as vertical bars. Lowercase letters above columns are used to present results of statistical data analysis obtained by Fisher's LSD test and point to statistically significant difference ($p < 0.05$) between values of $PI_{abs}$ measured at 3, 10, 18, and 26 days after flowering for each wheat variety separately and at each location.

3.2.2. Maximum Quantum Yield of Primary Photochemistry ($TR_0/ABS$) and Performance Index on Absorption Basis ($PI_{abs}$) in the Glumes of the Heads

At Osijek, variety El Nino significantly decreased $TR_0/ABS$ in the glumes of heads at the last measurement (26 DAF) compared to previous measurements (3, 10, and 18 DAF). In Tika Taka, the values of $TR_0/ABS$ remained unchanged through the whole measurement period. After an initial increase, Vulkan at Osijek significantly decreased $TR_0/ABS$ at the third measurement (18 DAF) compared to the second measurement (10 DAF), with a further decrease at 26 DAF. At Tovarnik, after an initial increase, El Nino and Vulkan significantly decreased $TR_0/ABS$ at the third measurement (18 DAF), after which a significant decrease occurred at the last measurement (26 DAF) compared to 18 DAF. Tika Taka at Tovarnik significantly decreased $TR_0/ABS$ at the last measurement (26 DAF) compared to previous measurements (3, 10, and 18 DAF) (Figure 6a–f).

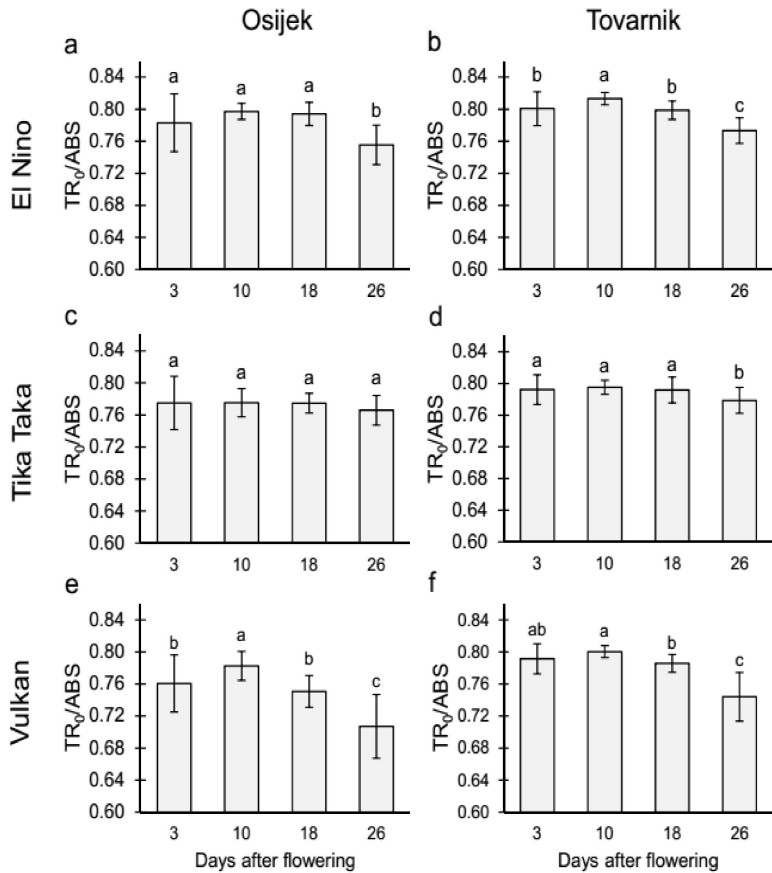

**Figure 6.** Maximum quantum yield of primary photochemistry (TRo/ABS) presented as mean value ± standard deviation of 20 wheat heads of variety El Nino (**a**,**b**), Tika Taka (**c**,**d**), and Vulkan (**e**,**f**) through four measurement points at two locations separately (Osijek and Tovarnik). Standard deviations are presented as vertical bars. Lower-case letters above columns are used to present results of statistical data analysis obtained by Fisher's LSD test and point to statistically significant difference ($p < 0.05$) between values of TRo/ABS measured at 3, 10, 18, and 26 days after flowering for each wheat variety separately and at each location.

$PI_{abs}$ was significantly decreased in the glumes of heads of El Nino at Osijek at 18 DAF, with a further decrease at 26 DAF. In Tika Taka, a decrease in $PI_{abs}$ already occurred at 10 DAF and continued to decrease at 18 DAF. After an initial increase at 10 DAF, Vulkan significantly decreased $PI_{abs}$ at 18 DAF, as well as at 26 DAF. At Tovarnik, all varieties significantly decreased $PI_{abs}$ at the second measurement (10 DAF), after which a significant decrease occurred at 18 DAF in El Nino and Vulkan. Values of $PI_{abs}$ in Tika Taka remained similar till 26 DAF (Figure 7a–f).

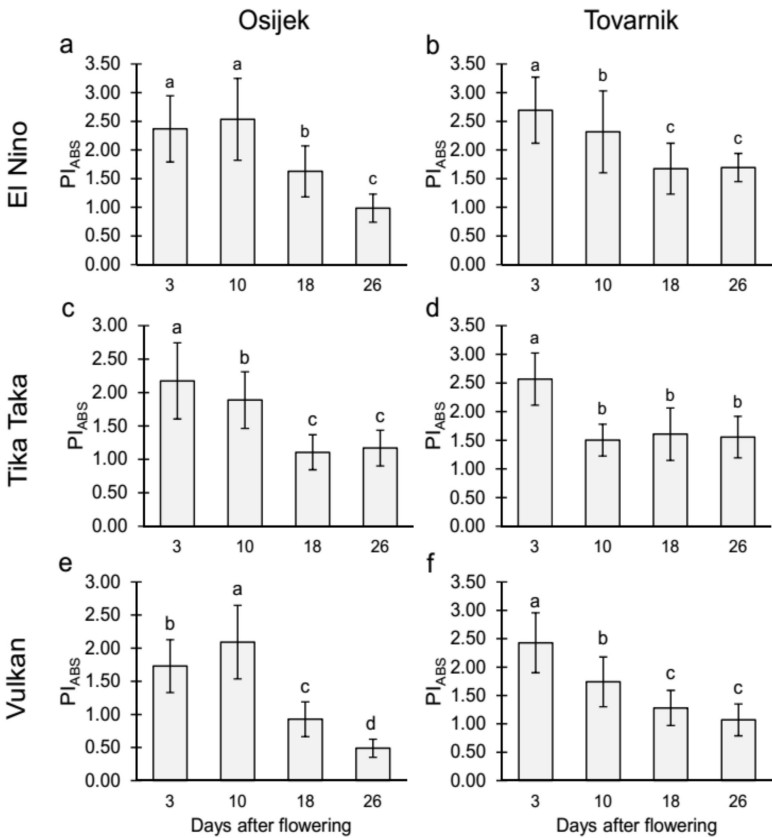

**Figure 7.** Performance index (PI$_{abs}$) presented as mean value $\pm$ standard deviation of 20 wheat heads of variety El Nino (**a**,**b**), Tika Taka (**c**,**d**), and Vulkan (**e**,**f**) through four measurement points at two locations separately (Osijek and Tovarnik). Standard deviations are presented as vertical bars. Lower-case letters above columns are used to present results of statistical data analysis obtained by Fisher's LSD test and point to statistically significant difference ($p < 0.05$) between values of PI$_{abs}$ measured at 3, 10, 18, and 26 days after flowering for each wheat variety separately and at each location.

## 4. Discussion

In the present study, grain yield and two agronomical components (test weight and 1000 kernel weight) were analyzed under two environmental conditions in three winter wheat varieties with a similar grain yield capacity, in parallel with the photosynthetic responses of their flag leaves and glumes of plant heads from flowering till the mid-grain-filling period. In the vegetative season 2019/2020, two field experiments were conducted at two different locations. Under natural field conditions without the use of fungicides, plants were exposed to disease stress factors, especially during the flowering stage. These could disrupt the photosynthetic apparatus, causing a decrease in plant productivity and overall grain yield.

### 4.1. Agronomical Traits

Significant differences in the average grain yields of winter wheat varieties were observed between the two locations. The grain yield of all varieties together at Osijek was significantly higher than at Tovarnik, although we could not undertake a statistical evaluation for each wheat variety separately, as field plots of 7.56 m$^2$ were replicated twice at each location. However, despite this, wheat producers and breeders believe the differences in grain yield between the two locations for each variety are huge. Significant differences on average for grain yield were expected, as wheat plants growing under field conditions were exposed to different environmental effects at those two locations. Previously, it was reported [32] that temperature and precipitation had a negative effect on

wheat yield. Furthermore, in a previous study [33], it was concluded that wheat yields can be severely reduced in stressful environments with a low rainfall amount. It was found that increases in precipitation led to a decrease in grain yield variability for wheat, maize, and cotton [34].

In the current study, the test weight was significantly lower at Osijek compared to Tovarnik. This was expected, as test weight is a density measurement that is used as an indication of grain quality but is not a factor in determining grain yield [35]. Due to that, large kernels have a lower test weight [36]. The kernel weight component is a significantly important determinant of grain yield because plant senescence overlaps with grain filling, while other yield components are largely determined before the initiation of the senescence. At both locations, 1000 kernel weight in Tika Taka was significantly larger than those in El Nino and Vulkan, reflecting the higher grain yield in Tika Taka at both locations on average (108.04 dt ha$^{-1}$). In some studies, varieties with delayed senescence increased kernel weight; therefore, yield also increased [37]. However, smaller kernel weight on average at Tovarnik suggests that the translocation of assimilates was not sufficient to fulfill the previously established grains in the early and mid-stage of grain filling. Furthermore, plants were under disease stress at Tovarnik due to higher precipitation favoring Fusarium head blight infection. It can be assumed that not all assimilates were incorporated into the grain, resulting in shriveled grains. Therefore, at Tovarnik, grain yield was reduced due to smaller grains and spikelet sterility in the lower and upper parts of the plant heads. This is in accordance with the research of Wollmer et al. (2016), where high precipitation at the time of grain filling shortened this phase and decreased grain yield due to smaller grains [3]. Previously, it was reported that rainfall patterns significantly reduced wheat yield [38]. In the current research, at Osijek, wheat varieties on average had significantly higher grain yield, which means that they could maintain a dynamical balance of photosynthetic products between photosynthetic sources (leaves) and nonphotosynthetic sinks (developing seed).

### 4.2. Chlorophyll (Chl) Fluorescence Parameters

Two chlorophyll (*Chl*) fluorescence parameters of flag leaves and heads in three wheat varieties were evaluated in the current study. Maximum quantum yield of primary photochemistry ($TR_0/ABS$) and performance index ($PI_{abs}$) parameters showed differences in the flag leaves and glumes of plant heads at the two locations during four measurement points. According to Lawlor (2002) [39], the performance of the photosynthetic process during grain growth can be affected by stomatal and nonstomatal limitations. In the research of Tsimilli-Michael and Strasser (2008), OJIP test parameters were used to investigate the vitality of plants and how they respond to different environmental conditions [40]. Some studies focus mainly on $TR_0/ABS$ and/or effective quantum yield of PSII [41]. However, biochemical limitations appear, mainly in the late stages of grain filling [42].

$TR_0/ABS$ is a sensitive indicator that, in stress, can indicate the downregulation of photosynthesis, and lower values of $TR_0/ABS$ may indicate stress and/or photoinhibition [43,44]. Along with that goes the fact that $TR_0/ABS$ can be quite insensitive to environmental changes [40]. It was reported that $TR_0/ABS$ is widely used as an indicator of the degree of plant stress [45]. The existence of any type of stress that results in inactivation or damage of PSII [46,47] results in decreased $TR_0/ABS$. In the current study, in the flag leaves, wheat varieties at the two locations reacted similarly considering $TR_0/ABS$ by decreasing it at the third (18 days after flowering, DAF) or last measurement points (26 DAF), which was expected as a consequence of senescence process in the flag leaves.

In the plant heads of the tested wheat varieties, $TR_0/ABS$ was different between the locations at different measurement points. This is an indication that PSII in the heads functioned differently at the different locations, which might be associated with FHB stress. At Osijek, variety Tika Taka did not show any stress considering $TR_0/ABS$, with the highest grain yield compared to other varieties. At Tovarnik, a decrease in $TR_0/ABS$ in Tika Taka occurred at 26 DAF. An earlier decrease in $TR_0/ABS$ also occurred for El

Nino. Vulkan decreased $TR_0/ABS$ at the same measurement point at both locations, which resulted in the lowest reduction of grain yield when comparing the two locations. Therefore, the reduction of $TR_0/ABS$ in the plant heads at Tovarnik was a good indicator of photosynthetic impairment resulting from FHB stress. This is in accordance with the research of Jing et al. (2009), who reported that $TR_0/ABS$ can be used as an indicator of the effects of environmental stress on photosynthesis [48].

Performance index on absorption basis ($PI_{abs}$) provides useful and quantitative information about the vitality of plants and combines the three main functional steps taking place in PSII (light energy absorption, excitation energy trapping, and conversion of excitation energy to electron transport) [49]. Additionally, it reflects the functionality of both PSI and PSII and gives quantitative information on the current state of plant performance under stress conditions [30]. Along with $TR_0/ABS$, *Chl* fluorescence parameter $PI_{abs}$ in the flag leaves of Vulkan decreased earlier compared to other varieties, thus giving the lowest yields at Osijek. In the wheat heads, all wheat varieties decreased $PI_{abs}$ earlier at Tovarnik compared to Osijek. Tika Taka decreased $PI_{abs}$ at the same measurement point at both locations. In the research of Fghire et al. (2015), $PI_{abs}$ was much more sensitive than $TR_0/ABS$, most probably because $TR_0/ABS$ responded to changes in the fast fluorescence rise kinetics between only the two fluorescence extremes, $F_0$ and $F_m$, while $PI_{abs}$ takes into account overall activity of both photosystems as well as the intersystem electron transport chain [22].

In general, it can be assumed that results from the current study showed an earlier decrease in $TR_0/ABS$ and $PI_{abs}$ in the plant heads, as well as an earlier decrease in $PI_{abs}$ in the flag leaves at Tovarnik compared to Osijek, most probably due to FHB disease stress at Tovarnik.

*4.3. Grain Productivity and Two Stress Indicators at Two Locations*

Three winter wheat varieties differing in grain yield at two locations were characterized by significant differences in the measured photosynthetic parameters of flag leaves and glumes of plant heads. In a previous study, it was concluded that the leaf senescence pattern is variable, and differences in leaf senescence should exist among varieties with differences in grain yield [50]. Measurements showed that $TR_0/ABS$ and $PI_{abs}$ in the flag leaves and glumes of plant heads were important to identify location differences. Tovarnik, where lower grain yield was obtained compared to Osijek, was characterized by an earlier decrease in values of $PI_{abs}$ in the heads and flag leaves. Furthermore, an earlier decrease in values of $TR_0/ABS$ in the heads at early and mid-grain filling occurred. These results indicated that decreased PSII activity, evaluated through chlorophyll fluorescence analysis, resulted in lower grain yields as a consequence of higher Fusarium head blight pressure due to increased precipitation. This is supported by the research of Frankel (1976), who reported that the sensitive period for each floret was detected between the initiation of the subtending lemma and the initiation of the floret itself [51]. Furthermore, it was concluded that the stage of flowering in wheat during the optimal period is critical for grain yield, as grain number is determined just prior to and at flowering, and grain yield is the most sensitive to stresses during this period [52]. A drastic decline in the activities of PSII and PSI indicated that photochemical activity inhibits photosynthesis during leaf senescence [53].

Delayed leaf senescence (stay-green trait) has long been considered to be a desirable trait in wheat breeding, as the divergence in grain yield is mostly the result of variation in the duration of photosynthetic activity. A recent study conducted on field-grown wheat demonstrated that photosynthetic traits (leaf gas exchange and chlorophyll *a* fluorescence) were positively correlated with the grain and harvest index [54]. Previously, a positive correlation between the stay-green trait, $TR_0/ABS$, and grain yield in field conditions was reported [55]. In the current research, wheat varieties at Osijek compared to Tovarnik displayed less inactivation of the PSII reaction center in the plant heads, which helped to maintain normal levels of PSII photochemical activity and photosynthetic electron transport.

## 5. Conclusions

At the Tovarnik location, higher Fusarium head blight (FHB) disease pressure occurred due to an unfavorable amount of precipitation during early and mid-grain filling. This caused an earlier decrease in the maximum quantum yield of primary photochemistry ($TR_0/ABS$) and performance index ($PI_{abs}$) in the glumes of plant heads, together with an earlier decrease in $PI_{abs}$ in the flag leaves at Tovarnik compared to Osijek. Our results suggested that decreased photosynthetic efficiency could shorten the grain-filling period in wheat varieties at Tovarnik, where inadequate assimilate supply around flowering period resulted in spikelet sterility through a fault in the pollination process. Additionally, it can be concluded that environmental stress in the stage of early grain filling influenced spikelet sterility and lowered 1000 kernel weight, thus resulting in lower final grain yield at Tovarnik in comparison to Osijek.

**Author Contributions:** Conceptualization, V.S.; methodology, V.S. and Z.K.; Z.K. and S.M.; formal analysis, V.S., S.M., Z.Z. and Z.K.; investigation, Z.Z.; resources, V.S.; writing—original draft preparation, V.S.; writing—review and editing, S.M., Z.Z. and Z.K.; project administration, V.S. and Z.Z. All authors have read and agreed to the published version of the manuscript.

**Funding:** This research was co-financed by the European Union through the European Regional Development Fund-the Competitiveness and Cohesion Operational Programme for grant number KK.01.1.1.04.0067.

**Institutional Review Board Statement:** Not applicable.

**Informed Consent Statement:** Not applicable.

**Conflicts of Interest:** The authors declare no conflict of interest.

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
