# Peer review of "Field Study of the Effects of Two Different Environmental Conditions on Wheat Productivity and Chlorophyll Fluorescence Induction (OJIP) Parameters"

_agriculture, doi:10.3390/agriculture11111154_

Round 1

Reviewer 1 Report

The authors have improved this manuscript considerably since first submission and the figures are clearer and understandable

Minor:

Please define OJIP test parameters at first insertion in the text i.e Chlorophyll fluorescence induction (OJIP) parameters

EDIT for line 368

although we could not undertake a statistical evaluation for each wheat variety separately, as field plots of 7.56 m2 were replicated twice at each location. However, despite this wheat producers and breeders believe the differences in grain yield between two locations for each variety is huge.

Ref 11: there are a lot of paper around the impact of  ear photosynthesis (see two below, but there are others), also values are different depending on wheat variety and climate - Please update text

The contribution of ear photosynthesis to grain filling in bread ...by ML Maydup2010Cited by 179 — In summary, ear photosynthesis makes a significant contribution to grain yield of bread wheat, from 13% to 33% in the absence of stress

Contribution of ear photosynthesis to grain yield under rainfed ...by Y WANG2016Cited by 38 — Grain yield, yield components, and ratio of grain weight:leaf area were positively related with contribution of ear photosynthesis

Ref 41 add following reference

Chlorophyll fluorescence analysis: a guide to good practice and understanding some new applications. EH Murchie, T Lawson. Journal of experimental botany 64 (13), 3983-3998

Ref 43. The paper on photoinhibition by Steve is quite old now. 1994. There are a number of recent reviews that could be included here.

Author Response

Dear reviewer 1,

Thank you very much for your comments. We corrected the manuscript according to your suggestions.

Minor:

Please define OJIP test parameters at first insertion in the text i.e Chlorophyll fluorescence induction (OJIP) parameters

*We defined OJIP test parameters when first time mentioned.

EDIT for line 368

although we could not undertake a statistical evaluation for each wheat variety separately, as field plots of 7.56 m2 were replicated twice at each location. However, despite this wheat producers and breeders believe the differences in grain yield between two locations for each variety is huge.

*We edited this line as you recommended.

Ref 11: there are a lot of paper around the impact of  ear photosynthesis (see two below, but there are others), also values are different depending on wheat variety and climate - Please update text

The contribution of ear photosynthesis to grain filling in bread ...by ML Maydup · 2010 · Cited by 179 — In summary, ear photosynthesis makes a significant contribution to grain yield of bread wheat, from 13% to 33% in the absence of stress

Contribution of ear photosynthesis to grain yield under rainfed ...by Y WANG · 2016 · Cited by 38 — Grain yield, yield components, and ratio of grain weight:leaf area were positively related with contribution of ear photosynthesis.

*We inserted those two references in the manuscript as you recommended.

Ref 41 add following reference

Chlorophyll fluorescence analysis: a guide to good practice and understanding some new applications. EH Murchie, T Lawson. Journal of experimental botany 64 (13), 3983-3998

*We added this reference in the text.

Ref 43. The paper on photoinhibition by Steve is quite old now. 1994. There are a number of recent reviews that could be included here.

*We added new reference in the text.

Reviewer 2 Report

In their resubmitted manuscript “Field study of the effects of two different environmental conditions on wheat productivity and OJIP test parameters” Spanic et al. describe their findings of the comparison of three different winter wheat varieties on two different locations. They analysed their performance and measured a set of parameters that give an insight into their e. g. yield.

They specifically focus their measurements on the flag leaves of the analysed plants and relate their data with the environmental settings the plants grew in.

In comparison to the first submission the quality of the manuscript increased not only in the presentation of the data but also in the language. Non the less in my opinion the text would still benefit if a native speaker would have a look at it to increase the readability.

Spanic et al. implemented all optimizations in their manuscript that were suggested in the reviews in the initial submission thus this version of the manuscript is easier to read and understand to the reader.

Author Response

Dear reviewer 2,

Thank you for your valuable comments, teacher of English went through the manuscript and re-checked the manuscript again.